# A prospective, multi-site, cohort study to estimate incidence of infection and disease due to Lassa fever virus in West African countries (the *Enable* Lassa research programme)–Study protocol

Suzanne Penfold[1]*, Ayola Akim Adegnika[2,3,4], Danny Asogun[5], Olufemi Ayodeji[6], Benedict N. Azuogu[7], William A. Fischer, II[8], Robert F. Garry[9], Donald Samuel Grant[10], Christian Happi[11], Magassouba N'Faly[12], Adebola Olayinka[13], Robert Samuels[10], Jefferson Sibley[14], David A. Wohl[8], Manfred Accrombessi[2], Ifedayo Adetifa[13], Giuditta Annibaldis[15], Anton Camacho[16], Chioma Dan-Nwafor[13], Akpénè Ruth Esperencia Deha[2], Jean DeMarco[8], Sophie Duraffour[15], Augustine Goba[10], Rebecca Grais[16], Stephan Günther[15], Énagnon Junior Juvénal Prince Honvou[2], Chikwe Ihekweazu[13], Christine Jacobsen[15], Lansana Kanneh[10], Mambu Momoh[10], Aminata Ndiaye[16], Robert Nsaibirni[16], Sylvanus Okogbenin[5], Chinwe Ochu[13], Ephraim Ogbaini[5], Énagnon Parsifal Marie Alexandre Logbo[2], John Demby Sandi[10], John S. Schieffelin[9], Thomas Verstraeten[1], Nathalie J. Vielle[15], Anges Yadouleton[2], Emmanuel Koffi Yovo[2], on behalf of the *Enable* Protocol authorship group¶

1 P95 Epidemiology and Pharmacovigilance, Leuven, Belgium, 2 Fondation pour la Recherche Scientifique (FORS), Cotonou, Bénin, 3 Centre de Recherches Médicales de Lambaréné, Lambaréné, Gabon, 4 Institut für Tropenmedizin, Universität Tübingen and German Center for Infection Research, Tübingen, Germany, 5 Institute of Lassa Fever Research and Control, Irrua Specialist Teaching Hospital, Irrua, Edo State, Nigeria, 6 Owo Federal Medical Centre, Owo, Ondo State, Nigeria, 7 Alex Ekwueme Federal University Teaching Hospital Abakaliki, Abakaliki, Ebonyi State, Nigeria, 8 Institute of Global Health and Infectious Diseases, The University of North Carolina (UNC) at Chapel Hill, Chapel Hill, NC, United States of America, 9 Tulane University School of Medicine, New Orleans, Louisiana, United States of America, 10 Kenema Government Hospital (KGH), Kenema, Sierra Leone, 11 Redeemer's University Nigeria, Ede, Osun State, Nigeria, 12 Université Gamal Abdel Nasser de Conakry, Conakry, Guinea, 13 Nigeria Centre for Disease Control, Abuja, Nigeria, 14 Phebe Hospital, Phebe, Liberia, 15 Department of Virology, Bernhard Nocht Institute for Tropical Medicine, Hamburg, Germany, 16 Epicentre, Paris, France

☯ These authors contributed equally to this work.
¶ Membership of the *Enable* Protocol authorship group is provided in the Acknowledgments.
* suzanne.penfold@p-95.com

**Data Availability Statement:** At the end of the study, the de-identified dataset underlying the

## Abstract

### Background

Lassa fever (LF), a haemorrhagic illness caused by the Lassa fever virus (LASV), is endemic in West Africa and causes 5000 fatalities every year. The true prevalence and incidence rates of LF are unknown as infections are often asymptomatic, clinical presentations are varied, and surveillance systems are not robust. The aim of the *Enable* Lassa research programme is to estimate the incidences of LASV infection and LF disease in five West African countries. The core protocol described here harmonises key study components, such as eligibility criteria, case definitions, outcome measures, and laboratory tests, which will maximise the comparability of data for between-country analyses.

findings will be available on request, in accordance with the legal framework set forth by Epicentre's data sharing policy, which ensures that data will be available upon request to interested researchers while addressing all security, legal, and ethical concerns. For data access, all readers may contact the Data Protection and Compliance Officer at dpco.archive@epicentre.msf.org.

**Funding:** The Coalition for Epidemic Preparedness Innovations (CEPI), an innovative global partnership between public, private, philanthropic, and civil society organisations to develop vaccines to stop future epidemics (https://cepi.net/), funds the Enable Lassa research programme. AAA (grant number PRJ-6086), WAF (PRJ-6087), RFG (PRJ-6088), MN (PRJ-6093), AO (PRJ-6089), and DAW (PRJ-6087) are the named direct recipients of funding from CEPI to implement the Enable Lassa research programme. The funders provide support in the form of salaries for AAB and HM. AAB and HM reviewed the study design and manuscript but have not had, and will not have, a role in data collection and analysis or the decision to publish.

**Competing interests:** The authors have declared that no competing interests exist.

## Method

We are conducting a prospective cohort study in Benin, Guinea, Liberia, Nigeria (three sites), and Sierra Leone from 2020 to 2023, with 24 months of follow-up. Each site will assess the incidence of LASV infection, LF disease, or both. When both incidences are assessed the LASV cohort ($n_{min}$ = 1000 per site) will be drawn from the LF cohort ($n_{min}$ = 5000 per site). During recruitment participants will complete questionnaires on household composition, socioeconomic status, demographic characteristics, and LF history, and blood samples will be collected to determine IgG LASV serostatus. LF disease cohort participants will be contacted biweekly to identify acute febrile cases, from whom blood samples will be drawn to test for active LASV infection using RT-PCR. Symptom and treatment data will be abstracted from medical records of LF cases. LF survivors will be followed up after four months to assess sequelae, specifically sensorineural hearing loss. LASV infection cohort participants will be asked for a blood sample every six months to assess LASV serostatus (IgG and IgM).

## Discussion

Data on LASV infection and LF disease incidence in West Africa from this research programme will determine the feasibility of future Phase IIb or III clinical trials for LF vaccine candidates.

## Introduction

Lassa fever (LF) is a zoonotic disease caused by the Lassa fever virus (LASV) belonging to the *Arenaviridae* family [1]. The most common mammal reservoir is the multimammate rodent, *Mastomys natalensis* [2]. LASV is transmitted to humans via direct contact with or ingestion of items contaminated with infected rodent excreta. Person-to-person transmission also occurs through contact with infected blood or body secretions in healthcare settings [3]. LASV infections most commonly occur in resource-poor, rural areas. Factors that may increase the risk of transmission through increasing human-rat contact include poor sanitation, crowding, deforestation, rodent hunting, bush burning, civil unrest, and the practice of drying grains by roadsides or outside homes and unprotected grain storage within homes [4].

LASV infection is asymptomatic in approximately 80% of the cases [3]. Although LF is considered a haemorrhagic illness, symptoms of LF are varied, non-specific and often difficult to distinguish from other febrile illnesses (e.g., malaria, typhoid) [5]. In symptomatic cases, headache, fever, muscle/joint pain, diarrhoea, vomiting, and elevated liver enzymes and haematocrit can appear 1–3 weeks after exposure to the virus. Facial oedema, shock, seizures, tremor, disorientation, severe haemorrhagic fever with multi-organ failure and coma may be seen in the later stages of disease. A third of the survivors suffer from sensorineural hearing loss (SNHL) as sequelae [6]. Approximately 1% of those infected with LASV die and the case fatality risk (CFR) increases to 20% among the hospitalised cases. The disease is severe in late pregnancy and can result in maternal death or miscarriages in 65% of cases [7, 8]. Once LF is suspected, interventions include isolating the patients in healthcare facilities to prevent further transmission and following strict infection control guidelines such as using personal protective equipment. The only specific treatment currently available is an antiviral drug, ribavirin [9]. However, the benefits have not been well evaluated [9]. There are currently no vaccines available.

LF is considered endemic in several countries in West Africa and is assumed to affect approximately 100,000 to 300,000 people, leading to 5,000 fatalities every year [10]. Nigeria has been reporting LF outbreaks almost every year, and annually since 2007 (A. Olayinka, personal communication, 16th July 2022). Studies have shown the LASV prevalence among Nigerian population ranged from 12–42% [11]. A study in Guinea reported seroprevalence of Lassa IgG to be 40.3% and IgM to be 2.8% [12]. Similar high LASV seroprevalence has been reported in Sierra Leone (8–40%) [11, 13]. Since 2014, Benin has been experiencing LF outbreaks almost every year, predominantly in the northern regions of the country, but no data exist on the incidence of the disease in Benin (A. Akim Adegnika, personal communication, 1st August 2022). However, the true prevalence and incidence of LF are not known as many people are asymptomatic, many present with a non-specific and wide spectrum of clinical presentations, and robust surveillance systems are lacking in some countries [6]. Furthermore, the World Health Organization (WHO) has identified several remaining gaps in the understanding of LF including the contribution of viral shedding in asymptomatic individuals to human-human transmission; the pathogenesis and immunology of LASV infections to support the development of diagnostics, therapeutics and vaccines against LF; the determinants of LASV infection and disease severity; and descriptive epidemiological data on LF incidence and LASV seroprevalence by clade, geographic area and other population demographics [14]. Therefore, the aims of the *Enable* programme are to assess LF disease and LASV infection incidence in West African countries (Benin, Guinea, Liberia, Nigeria, and Sierra Leone) and to determine the feasibility of future late-stage clinical trials for assessing the efficacy of LF vaccine candidates. The programme will also help to strengthen site and investigator capacity to perform future vaccine trials.

The primary objectives of the *Enable* programme are:

1. To assess the incidence rate of symptomatic confirmed LF separately for each participating Lassa-endemic site.

2. To estimate the incidence rate of LASV infection separately for each participating Lassa-endemic site.

The secondary study objectives, to be assessed across the *Enable* programme and in each study site, are listed in Table 1. The outcome measures for the objectives are detailed in Table 2.

This paper describes the core study protocol that was developed through contributions from each study site team, implementing partners and other technical experts. The core protocol was subsequently adapted by each study site team for implementation according to their setting, resources and planned staffing.

## Materials and methods

### Study design, setting and study period

The *Enable* programme consists of seven prospective cohort studies that are being conducted in LF endemic countries in West Africa; Benin, Guinea, Liberia, Nigeria, and Sierra Leone, from late 2020 to the end of 2023. Participants will be followed for 12 to 24 months.

The study consists of two components: i) an LF disease cohort to assess the incidence of symptomatic LF cases and ii) a LASV infection cohort to estimate the incidence of LASV infection. Study sites either implement both components or the infection cohort only (Table 3). Sites implementing both components draw the (smaller) LASV infection cohort as a nested cohort from the (larger, main) LF disease cohort (Fig 1).

Table 1. Secondary objectives of the *Enable* Lassa research programme.

| | Secondary Objectives |
|---|---|
| **Related to primary objective 1** | |
| 1 | To assess the overall and age-specific incidence rate of confirmed LF |
| 2 | To assess the LASV clade-specific incidence rate of confirmed LF |
| 3 | To assess the association between baseline seropositivity and the occurrence of confirmed LF |
| 4 | To describe the clinical course of LF, specifically among risk groups (pregnant women, children, and the elderly) |
| 5 | To determine the CFR among confirmed LF |
| 6 | To determine the proportion of confirmed LF patients with persistent Sensorineural Hearing Loss (SNHL) |
| 7 | To determine the proportion of LF survivors with delayed or persistent SNHL |
| 8 | To assess the incidence rates of co-infection of malaria in confirmed LF cases |
| 9 | To assess the role of selected risk factors (age, sex, occupation, housing characteristics, previous LF diagnosis, contact with rodents, presence of rodents in the house) for symptomatic confirmed LF disease |
| 10 | To determine the proportion of confirmed LF among acute febrile illness cases |
| 11 | To define various levels of severity of symptomatic confirmed LF cases |
| **Related to primary objective 2** | |
| 1 | To assess the overall and age-specific LASV seropositivity prevalence at baseline |
| 2 | To estimate the overall and age-specific incidence rate of LASV infection (seroconversion) |
| 3 | To assess the age-specific prevalence of seropositivity at baseline and over time |
| 4 | To assess the role of selected risk factors (age, sex, occupation, housing characteristics, previous LF diagnosis, contact with rodents, presence of rodents in the house) for LASV infection at baseline and over time |
| 5 | To assess the overall and age-specific incidence of seroreversion |

## Study governance and roles of various partners

The *Enable* programme is implemented by several partners (Table 4) collaborating in an agreed governance structure (S1 Fig). In each site, the study is implemented by a team comprising field and laboratory researchers, data managers, administrative support, and clinicians. Each team is headed by principal investigators, at least one of whom is a study country national.

The *Enable* programme is funded and facilitated by the Coalition for Epidemic Preparedness Innovation (CEPI). Specifically, CEPI is responsible for overall programme monitoring and evaluation, financial oversight (contract/grant management and financial audits), contracting and centralised procurement, and policies and procedure development. The programme is further coordinated by a group of four partners who, together with CEPI, form the programme headquarters (PHQ). P-95 is the programme coordinating partner responsible for programme management, regular monitoring of programme finances, logistics and delivery support, training support, field monitoring, implementation of policies and procedures, and reporting to and implementing decisions made by the Programme Steering Committee (PSC). Margan Clinical Research Organization (MMARCRO) conducts site monitoring and overall quality management. Epicentre is responsible for data management, including platform development, training, and support; development of the data management plan (DMP) and statistical analysis plan (SAP); and conduct of cross-country analyses. The Bernhard Nocht Institut for Tropical Medicine (BNITM) provides laboratory support in the form of standard operating procedure harmonisation, laboratory staff training, and field support.

Governance of the study among all partners is ensured by the following groups:

1. The PSC, comprised of one voting member from each study country (with an annually rotating chair), provides overall governance for the programme to ensure the successful

**Table 2. Outcome measures of the *Enable* Lassa research programme.**

| Cohort | Outcome Measures | Definition/Description |
|---|---|---|
| **Disease** | Incidence rate of acute febrile illness | Number of acute febrile illness episodes per 1,000 person-years of follow-up in the symptomatic disease cohort, overall and stratified by country and site. An acute febrile illness case is defined as a self-reported fever of > 48 hours duration (lasting at least two consecutive nights) + 1 of the signs/symptoms- headache, abnormal bleeding (from mouth, nose, rectum, and/or vagina), chest pain, oedema of the neck/face, joint pain, conjunctival or sub-conjunctival haemorrhage, vomiting, jaundice, cough, spontaneous abortion, sore throat, buzzing in ears/acute deafness, abdominal pain, hypotension, and recent contact with a confirmed LF case. |
| | Incidence rate of symptomatic confirmed LF | Number of confirmed LF cases per 1,000 person-years of follow-up overall and stratified by country, site, age groups, gender, viral clade, and baseline serostatus. A confirmed LF case is someone fulfilling an acute febrile illness case definition and has a positive Lassa RT-PCR test result. |
| | Clinical course of LF | Percentage of confirmed LF cases classified with a disease stage 3 or 4 (see SAP) at different timepoints (admission, during admission and before discharge) and stratified by risk groups |
| | Case fatality risk | Percentage of confirmed LF cases dying within 30 days of diagnosis or attributable to LF disease as assessed by the treating clinician at any point past confirmation |
| | Occurrence of SNHL | Percentage of all confirmed LF cases with SNHL assessed by audiometry prior to discharge. SNHL is defined as hearing loss of at least 30dB in three sequential frequencies in the standard pure tone audiogram, where a physical examination has excluded conductive hearing loss. |
| | Occurrence or delayed/persistent SNHL/ other sequelae | Percentage of all LF survivors with SNHL at 4 months after hospital discharge. Delayed SNHL is defined as audiometry consistent with SNHL at follow-up but not at hospitalisation. Persistent SNHL is defined as audiometry consistent with SNHL at hospitalisation and at follow-up. |
| | Prevalence of symptomatic confirmed LF co-infected with malaria parasites | The percentage of all confirmed LF cases among whom presence of malaria parasites assessed by antigen RDT is detected at the time of LF diagnosis |
| | Risk factors for symptomatic confirmed LF | Association between the incidence rate of symptomatic LF disease and prespecified characteristics of the study subjects, expressed as an incidence rate ratio |
| | LF Severity | Due to the absence of a well-established severity scoring system, a severity scoring system for use in clinical trials will be developed from clinical and laboratory information collected from confirmed LF cases. |
| **Infection** | Incidence rate of LASV infection | The number of individuals who change from seronegative (absence of Lassa-specific serum antibodies in either IgG or IgM-ELISA) to seropositive status (presence of Lassa-specific serum antibodies in either IgG or IgM-ELISA) per 1,000 person-years of follow-up in the infection cohort, overall and stratified by country, site, and age groups |
| | Seroprevalence | The percentage of subjects found to be seropositive (IgG+) out of the number of subjects tested for baseline seropositivity overall and stratified by country, site, and age group |
| | Risk factors for LASV infection | The association between the incidence rate of LASV infection and prespecified characteristics of the study participants or their households, expressed as an IRR |

(*Continued*)

**Table 2.** (Continued)

| Cohort | Outcome Measures | Definition/Description |
|---|---|---|
| | Risk factors for seropositivity: | The ratio of the odds of being seropositive at baseline in prespecified risk groups to the odds in the study population without the prespecified risk, expressed as an odds ratio (OR) |
| | Incidence of seroreversion | The number of individuals who change from seropositive to seronegative per 1,000 person-years of follow-up in the infection cohort, overall and stratified by country, site, and age groups |

implementation of the study protocol across multiple sites. The PSC meets quarterly. PSC meetings are also attended by the chair of the Epidemiology Expert Reference Group (EERG) and the Council of Public Health Authorities (CPHA), a WHO representative, and members of the CEPI project team and PHQ as appropriate. PSC meetings are hosted by CEPI and chaired by the PSC Chair.

2. The CPHA (Launch of the CPHA has been delayed due to the COVID-19 pandemic. In the interim, each country implementing partner has engaged directly with the appropriate senior public health authority and other key stakeholders in their country to ensure visibility of *Enable* programme activities) is an advisory body to the PSC composed of representatives from the relevant public health authority in each study country. Its role is to provide guidance and ensure alignment of study activities with country research strategies and needs.

3. The EERG is an advisory body to the PSC composed of Lassa virus, epidemiology, vaccinology, and public health subject matter experts. Its role is to provide independent scientific expert advice on the design and implementation of the *Enable* programme.

During the development of the protocol, technical working groups were formed as a forum for discussion and knowledge sharing in key technical areas (including field implementation, laboratory analysis, data management, capacity strengthening, and communication and community engagement). Each group comprised leaders and subject matter experts from each study country, independent experts, and CEPI and PHQ staff.

### Selection criteria

**Site selection criteria.** Within each country, the study will be conducted at localities where available data suggested that the risk of LASV infection is relatively high, an assessment

**Table 3. Countries and sites where the study is being conducted and the cohorts being conducted in each.**

| Site number | Country | State | Major community/ies | Cohort(s) being conducted |
|---|---|---|---|---|
| 1 | Nigeria | Edo | Etsako, Esan northeast and Esan southwest Local Government Areas (LGAs) | LF Disease and LASV infection |
| 2 | Nigeria | Ondo | Akure North, Owo and Ose LGAs | LASV Infection |
| 3 | Nigeria | Ebonyi | Abakaliki, Ebonyi and Izzi LGAs | LASV Infection |
| 4 | Liberia | Not applicable | Bong county | LF Disease and LASV infection |
| 5 | Sierra Leone | Not applicable | Kenema, Bo, Kono and Kailahun districts | LF Disease and LASV infection |
| 6 | Guinea | Not applicable | Faranah prefecture | LASV Infection |
| 7 | Benin | Not applicable | Tchaourou, Parakou and Natitingou communes | LF Disease and LASV infection |

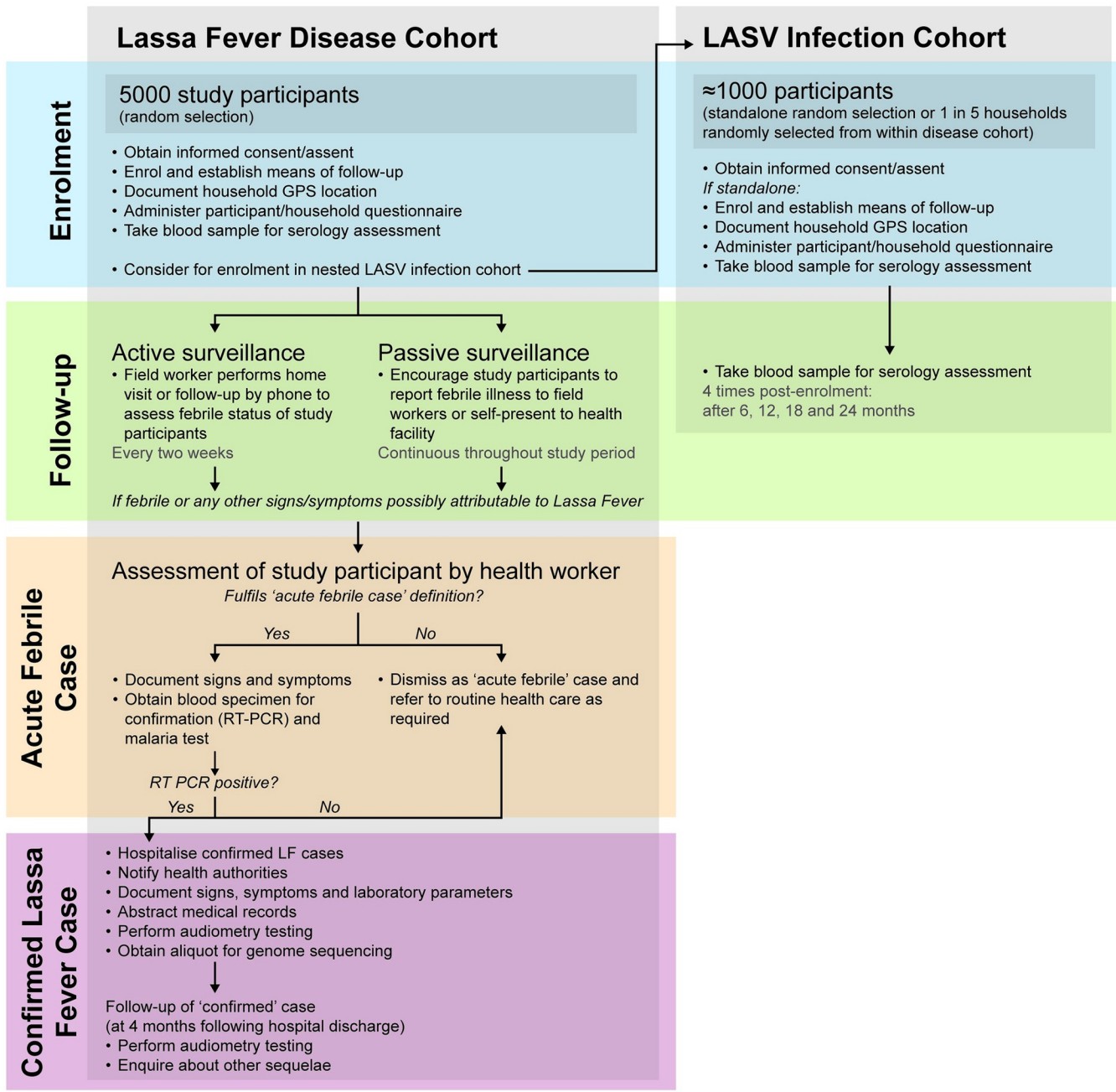

**Fig 1. Study flow diagram.**

study indicated they have capacity to perform future clinical trials, and the populations are relatively stable, with little expected inward or outward migration (Table 3 and Fig 2) [15]. The study localities are in well-defined geographical areas, including multiple rural and peri-urban localities, with healthcare facilities. Moreover, the localities are within the catchment area of referral-level healthcare facilities. These referral healthcare facilities have appropriate diagnostics infrastructure for LASV testing or, if diagnostic facilities are unavailable, the samples will be sent to national reference laboratory.

**Table 4. List of *Enable* Lassa research programme partners by country.** Partner organisations are listed in alphabetical order per country.

| Country | Partner organisation |
| --- | --- |
| Belgium | P95* |
| Benin | Ministère de la Santé, République du Bénin (inc. National Reference Lab, Cotonou) |
| | Conseil National de Lutte contre le VIH/Sida, la Tuberculose, le Paludisme, les Infections Sexuellement Transmissibles et les Épidémies (CNLS-TP) |
| | Foundation pour la Rechercher Scientifique (FORS) |
| | Institut de Recherche Clinique du Bénin (IRCB) |
| | University of Parakou |
| Canada | Université Laval |
| France | Epicentre* |
| Gabon | Centre de Recherches Médicales de Lambaréné |
| Germany | Bernard Nocht Institute for Tropical Medicine (BNITM)* |
| | Institute for Tropical Medicine, University of Tübingen (ITM-EKUT) |
| | Robert Koch Institute (RKI) |
| Ghana | Margan Clinical Research Organization (MMARCRO)* |
| Guinea | Faranah Regional Hospital |
| | Université Gamal Abdel Nasser de Conakry (UGANC) |
| Liberia | National Institute of Public Health, Liberia (NPHIL) |
| | Phebe Hospital |
| Nigeria | African Field Epidemiology Network (AFENET) |
| | Alex Ekwueme Federal University Teaching Hospital Abakaliki (AEFUTHA) |
| | Federal Medical Center–Owo (FMCO) |
| | Irrua Specialist Teaching Hospital (ISTH) |
| | Nigeria Centre for Disease Control (NCDC) |
| | NCDC National Reference Lab |
| | Redeemer's University (RUN)* |
| Norway | Coalition for Epidemic Preparedness Innovations (CEPI)* |
| Sierra Leone | Kenema Government Hospital |
| UK | UK Public Health and Rapid Support Team (UK-PHRST) |
| USA | International AIDS Vaccine Initiative (IAVI)* |
| | Tulane University* |
| | University of North Carolina (UNC) |

* Indicates a partner with activities across multiple countries. They are listed under the country where their head office/headquarters are located.

## Household and participant selection

Each study team will be responsible for identifying the study population based on epidemiological and logistic considerations (Table 3). To facilitate the acceptance of the study by the population, recruitment will be done at household level rather than at individual level. Depending on the study population in each locality, a stratified or two-stage cluster sampling method will be used for the random selection of locations and households to be included in the study (Table 5). The selection of households for the infection cohort within those identified for the disease cohort will be done by systematic random sampling.

**Participant inclusion and exclusion criteria.** Residents of the selected households will be eligible to participate if, at the time of recruitment, they are > = 2 years old who have been residents in the locality for six months preceding recruitment, not febrile, and expect to stay in

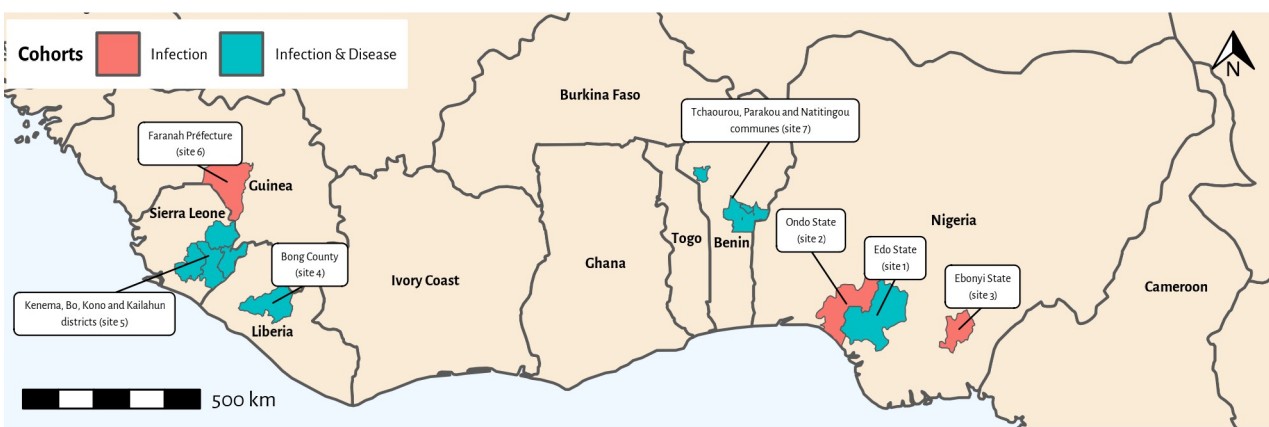

**Fig 2. Map of study sites.**

the area for most of the time until the end of the study period. The rationale for excluding children less than two years old is because local clinical experts reported LF to be rare in children under two years old, and there is low community acceptability of drawing blood from very young children. In addition, participants will need to be able to understand the official languages in which information about the study is available or have a surrogate available who can translate and suitable measures for future contact via the household head or secondary household contact person (i.e. telephone number) will be required.

Household members will not be recruited if they are unwilling to comply with any of the study procedures, including blood specimen collection, are indicated by the respective household head as not eligible, or for medical or social reasons, including fever at the time of recruitment. Furthermore, members will not be eligible to participate if they are not able to consent freely (e.g. military personnel), if there is any other reason at the discretion of the study personnel that would interfere with a person's ability to participate (e.g. mental illness), or if they are study staff or study personnel.

## Pre-study activities

Sensitisation of healthcare workers (HCWs) and communities will be planned and implemented in each study team according to the needs and setting. Broadly, sensitisation activities will include the following:

**Key stakeholders' involvement:** The study teams will identify key administrative and political authorities at national and subnational levels, including ministry of health authorities, traditional authorities, civil society and religious leaders, traditional healers, teachers, and medical staff at the community levels. In order to facilitate engagement with the study, they will hold meetings with these key individuals to explain the study, what is expected, and potential study outcomes, and to enable questions to be answered.

**Sensitisation of HCWs:** To prevent delayed case detection by the health staff at the health facility and to prevent nosocomial LF transmission, it will be critical to continuously sensitise HCWs on LF. Health authorities, as part of the study teams, will provide training and mentorship of HCWs according to WHO guidelines on LF case detection, diagnosis, and control to maintain a high LF index of suspicion, an adherence of HCWs to use protective measures while providing care, and on prompt treatment of diagnosed cases and supportive care.

**Sensitisation of the communities:** Various programmes will be conducted to sensitise the community. Mass media will be used to sensitise the communities about the study. At

**Table 5. Sampling methods being used in each site.**

| Country—site | Sampling of localities | Sampling of households within localities |
|---|---|---|
| **Benin** | We will randomly select a total of 80 villages in proportion to their population size, from a total of 103 villages in Tchaourou, Parakou and Natitingou districts. | To include about 63 participants (5000/80), approximately 12 households should be included in each village, based on available demographic data. We will record the boundaries of each village using GPS. We will use satellite imagery and a random selection algorithm to select 12 dwellings in each village, plus six dwellings in reserve. The dwellings will be visited consecutively using a smartphone equipped with a GPS application until approximately 63 (+/- 3) participants are included. If this number is not reached after visiting the 12 dwellings (e.g. dwelling unoccupied or refusal of the household), the dwellings from the reserve list will be visited. |
| **Guinea** | Thirty-two villages in the region of Faranah will be selected purposively by a commission based on four criteria: i) incidence of fever from an unknown origin higher than 4% ii) between 1000 and 2000 habitants iii) not located on main roads and iv) less than 45 from Faranah city by car (for logistic reasons). | We will calculate the number of participants required in each village in proportion to the size of the village. We will obtain an enumeration of households in each selected village from the previous household census conducted for bed net distribution. Within each village, we will select households by simple random sampling until the total number of eligible household members reaches the target number. |
| **Liberia** | We will purposively select three documented Lassa fever hotspot communities in Bong County. | We will calculate the number of participants required in each community in proportion to the size of the community and convert this into the number of households using available demographic data (average household size estimated at five members). We will record the boundaries of each community using GPS. We will use satellite imagery and a random selection algorithm to select the desired number of dwellings in each village, plus 50% in reserve. We will visit these dwellings consecutively using a smartphone equipped with a GPS application until the target number of participants is included, using the dwellings from the reserve list if necessary. |
| **Nigeria—Abakaliki** | We will select ten documented Lassa fever hotspot communities. | We will calculate the number of participants required in each community in proportion to the size of the community and convert this into number of households using available demographic data (average household size estimated at five members). We will conduct systematic random sampling from the exhaustive enumeration of all households in the community obtained from a previous immunization programme. We will continue recruitment until the target number of participants is reached. |
| **Nigeria–Irrua** | We will purposively select seven documented Lassa fever hotspot communities. | We will calculate the number of participants required in each community in proportion to the size of the community and convert this into the number of households using available demographic data (average household size estimated at five members). We will conduct systematic random sampling from the exhaustive enumeration of all households in the community obtained from a mini census. We will continue recruitment until the target number of participants is reached. |
| **Nigeria—Owo** | We will purposively select ten documented Lassa fever hotspot communities. | We will calculate the number of participants required in each community in proportion to the size of the community and convert this into number of households using available demographic data (average household size estimated at five members). We will conduct systematic random sampling from geographical data, including locations (streets), as well as demographic information on those geographical locations (number of households). We will continue recruitment until the target number of participants is reached. |
| **Sierra Leone** | We will randomly select 18 villages from the 3782 villages in four districts surrounding Kenema General Hospital, from where the number of Lassa cases have been previously admitted to the Lassa fever ward. The 3782 villages are distributed over 385 sections. To maximise representativeness, we will use a two-stage cluster sampling method: the first stage will be to sample the sections with probability proportional to size and the second stage will be to select randomly one village in each section. Owing to the geography of the villages (too far apart to be adequately grouped in the same cluster), we will not include settlements with fewer than 20 dwellings. | Given the small size of the villages (on average about 500 inhabitants) and to facilitate acceptance of the study, we will include all households in a village. We will enrol villages until the target population is reached. |

community meetings local LF survivors will be invited to share their experience, people will be advised on rodent control measures and will be educated about the risk of transmission and the benefits of early reporting, and community health workers and community advisory boards will be involved, wherever possible.

## Study procedures -LF disease cohort

**Study participant enrolment.** Researchers conducting recruitment will be study team members trained in the study procedures and consent seeking who speak the language(s) spoken in the community in which the approved information and informed consent forms (ICFs) will be available. For each selected household, researchers will approach the residents to identify the household head and summarise the study and its aims to them. If the household head is not present at the time of first contact, the researchers will return to the household within 24 hours for a second attempt. If the household head remains unavailable and does not nominate an alternative household member in their absence, the household will be dismissed for inclusion in the study and not replaced.

Once the household head gives permission for the research team to enter the dwelling, the team will check household members as potential participants against the eligibility criteria. The researchers will inform potential study participants, or their parent/legal guardian if the participant is under five or ten years old (depending on the country), about the study verbally and in writing using approved written ICFs or audio translations. Informed consent (and assent in case of minors) to participate will be sought from each individual or their parent/legal guardian and indicated either by signature or fingerprint if illiterate. For illiterate participants, a literate witness will be present during the consent process and will also sign the ICFs. The research team will give each participant a unique study identifier and identification card containing a barcode and their photo. A maximum of 18 participants per household will be recruited in order to limit the impact of very large households (e.g. polygamous families) on the representativeness of the participants. If there are more than 18 eligible residents, potential participants will be selected by simple random sampling before assessing eligibility and seeking consent. If any selected household member declines to participate, they will not be replaced by another member of the household. If one or more household members are not present at the time of recruitment, the household will be revisited.

Compensation for participant's time to provide responses and blood samples will be determined by each country according to the community norms. Compensation will be detailed in the country-specific protocols and receive Ethical Committee approval.

**Baseline data collection.** Data on the Global Positioning System (GPS) position and description of the household will be collected using REDCap (Research Electronic Data Capture) on 4G tablets. The research team will complete a questionnaire for each household through a combination of observed and household head-reported information. Data collected will include household composition; socioeconomic status of the household as indicated by dwelling structure and amenities, occupation of resident adults, and assets owned; and exposure to potential LF risk factors. Researchers will record participants' age, sex, education level, ethnicity, occupation, and LF history. Additional questions about LF knowledge and risk factors will be asked as deemed useful by the country teams.

**Baseline blood sample collection.** Trained phlebotomists will take 5 ml of blood by venepuncture from a peripheral vein of each participant at baseline. Samples will be transported to the laboratories and divided into two cryovials: one to evaluate LASV serostatus (IgG) at baseline and the second will be stored for future assessments. Based on recommendations from the Foundation for Innovative New Diagnostics (FIND), which performed comparative

evaluations of the available IgG NP immunoassays for LF, and after considering other criteria (e.g., assay performance, kit production capacity, and production and delivery times) through a formal review process, CEPI selected the ReLASV® Pan-Lassa Antigen ELISA Kit–RUO (Zalgen, Maryland, USA) [16] for all IgG ELISA assays.

All blood specimens for testing will be collected, processed, stored as per laboratory standard operating procedures, and shipped as per site and laboratory partner requirements.

**Follow-up of study participants.** Researchers will contact household heads or secondary contacts every two weeks (called active follow-up). The contact persons will be asked to report on the status of all household members enrolled in the study. These checks will include confirmation of domicile, presence of febrile episode and other clinical symptoms according to the acute febrile illness case definition in the previous two weeks (Table 1), any visits to a health facility, and a reminder to report to the local designated health facility and the research team in case of fever that persists for two nights or more.

In case of a missed follow-up, the researchers will make two more contact attempts within 48 hours including at least one home visit. In case of no contact, that active follow-up event will be documented as missing. If a household cannot be contacted for two or more consecutive biweekly events, the household will be considered as temporarily lost to follow-up. This means they will not be contacted by the research team again but can re-enter the study if they contact the study team. If the household contact reports that any study participants have or have had fever in the previous two weeks, the researcher will ask the household head if the individual has/had any of the symptoms listed in the acute febrile illness definition (Table 1).

In between active follow-up visits, study participants or their parent/legal guardian will also be encouraged to immediately report any febrile episode that has persisted for at least two consecutive nights to the research team (called passive follow-up).

**Procedures after classification as acute febrile illness case by a fieldworker.** Following the identification of a possible acute febrile case (Fig 1), a HCW will interview the participant within 24 hours to confirm the clinical signs and symptoms. Those participants confirmed as meeting the acute febrile case definition will be asked to give a 5 ml blood sample. Laboratory personnel will test the blood samples for LASV using the RealStar® Lassa Virus real-time reverse transcriptase polymerase chain reaction (RT-PCR) kit 2.0 (Altona diagnostics, Germany). The kit is highly sensitive and specific, targets both the S and L genome of the LF virus, and was validated using clinical samples from the study countries.

All implementing partners have access to laboratory facilities with RT-PCR testing capacity, either on-site or through collaboration with the national reference laboratory. Researchers will also test for the detection of Plasmodium falciparum–specific HRP2 antigens and common Plasmodium lactate dehydrogenase (pLDH) of Plasmodium species using the First Response® Malaria Antigen Card Test (Premier Medical Corporation Limited, India) or CareStart™ malaria HRP2/pLDH (Pf/pan) combo test (Access Bio, USA). The participant will be informed of the results.

*Follow-up of LF confirmed cases*. Once participants have been classified as confirmed LF cases, infection control and personal protection measures will be appropriately taken by the researchers who interact with them. The study team will inform the local authorities, who will then be responsible for their medical care and infection control measures. Researchers will extract data on clinical signs and symptoms, blood chemistry results, co-morbidities and medications and treatments given from the medical records of confirmed cases. Trained HCWs from the research team will conduct an audiometry test on each confirmed LF case before hospital discharge to assess for the presence of SNHL using Kuduwave model 5000 (Benin, Nigeria, and Liberia) or Shoebox professional (Sierra Leone) audiometers.

Aliquots obtained from all symptomatic LF cases will be sequenced. Redeemer's University in Nigeria will be responsible for LASV-positive samples collected in Nigeria, Benin, and Liberia using the Illumina platform. Kenema General Hospital in Sierra Leone will be responsible for sequencing samples collected in Sierra Leone, which will be carried out using the Oxford Nanopore (MinION) platform. Standard SOPs will be followed for sample handling, extraction, amplification, quantification, library preparation and sequencing. Bioinformatic analysis will be performed to define the type of specific LASV lineages circulating in the study countries.

LF survivors will be followed up four months after discharge. A trained HCW will visit the participant at home to perform a second audiometry test to assess for persistent or delayed onset SNHL. Furthermore, the participant will be asked about their health status and whether they are suffering from any sequelae. If sequelae persist, the HCW will refer the study participant for healthcare provision as appropriate.

**Recording of LF fatal cases.** Information on fatal LF confirmed cases will be obtained from medical records. In case the study team is informed of a study participant's death that occurred in the community, researchers will attempt to identify the cause of death via a verbal autopsy to be conducted with a family member up to four weeks following the participant's death.

## Study procedures- LASV infection cohort

**Study participant enrolment.** There are two scenarios (Fig 1) for participant selection for the LASV infection cohort:

**Scenario A:** The LASV infection cohort study component is implemented at the proposed sites in combination with the LF disease cohort study component. In this scenario, study participants for the (nested) LASV infection cohort will be selected through systematic sampling at the household level as they are enrolled in the (main) LF disease cohort until the targeted sample size is achieved. The potential LASV infection cohort study participant or their parent/guardian will be informed about the aims of the LASV infection cohort study component and its procedures. Additional informed consent (and assent as required) from the participant of their parent/guardian will be sought for the LASV infection cohort.

**Scenario B**: the LASV infection cohort study component is implemented by itself. In this scenario, study participants will be selected using a stratified sampling method at the household level described in Table 5. Informed consent (or assent) from the participant or their parent/guardian will be sought before enrolment.

**Baseline data and blood specimen collection.** For sites implementing scenario B, baseline data and blood specimen collection and tests will be performed as described for the disease cohort. Serological testing will be done by ELISA for IgG at baseline.

**Follow-up of study participants.** At 6, 12, 18, and 24 months after baseline, researchers will visit participants at home to draw another 5-ml blood sample. The sample will again be divided into two cryovials: one sample will be used to evaluate LASV serostatus by ELISA, this time testing for both IgG and IgM, and again the second will be stored for future assessments. The choice of kit selection will be based on a similar process as for the IgG kit, i.e. FIND will perform comparative evaluations of the available IgM NP immunoassays for LF and after considering other criteria (e.g., assay performance, kit production through a formal review process), CEPI will select the most appropriate kit.

## Data management

Data will be collected by research staff completing questionnaires and case report forms using REDCap (Research Electronic Data Capture) version 5.20.9 on password-protected tablets.

REDCap is a web-based software that allows offline data collection and is hosted within Epicentre's cloud infrastructure in Paris, France. Tablets will be synchronised daily with the central database over an encrypted internet connection. This infrastructure will guarantee the performance and availability of data collection and reporting tools. A separate database will be created for each study country and adequate access control mechanisms employed to provide sufficient visualisation and processing permissions to staff members on the study databases according to their role(s) in the study.

Hard copies of data collection forms and other study documentation (enrolment logs and sample collection/movement logs) will be available in case of issues with electronic data capture at the point of entry. Any data captured using paper forms will be entered retrospectively into the electronic data capture system. All changes or corrections to entered and stored data will be justified and documented in an audit trail.

Optimal data collection methods and a study quality control plan will be developed and implemented centrally. This plan will include the creation of secure automated dashboards with key performance indicators for data quality, study follow-up, and monitoring. A separate DMP (see supplement) will describe all functions, processes, and specifications for data collection, cleaning, extraction, and validation.

## Statistical analysis

All partners agreed upon a sample size of 5000 participants for the LF disease cohort and 1000 participants for the LASV infection cohort as it was the most reasonable balance between feasibility and desired precision for the range of expected incidence and seroprevalence rates. The size of the LF disease cohort enables to measure an incidence rate of symptomatic cases of 0.1% over the two-year period, with a precision from 0.04% to 0.23% (95% binomial confidence intervals calculated using Wilson method [17]). The size of the LASV infection cohort enables to measure a seroprevalence of LASV infection of 10% with precision from 8.3% to 12.0% (95% binomial confidence intervals calculated using Wilson method [17]). Precision levels for sample sizes between 200 to 25000, incidence of LF disease between 0,001% and 3% and seroprevalence of LASV infection between 1% and 50% are presented in the SAP (see supplement). The original sample size calculation in the protocol had several limitations: i) It could not account for the baseline LASV seroprevalence which is unknown in the study population. If the LASV seroprevalence in the study population is high, this is likely to reduce the number of people at risk in the cohort; ii) It did not take into account the clustering of participants within households and communities, which was necessary for feasibility and acceptance by the population; iii) It did not account for participant attrition during the 2-year follow-up period, which is likely to reduce the final cohort size for the analyses. As a result, the precision intervals specified in the protocol are likely to be narrower than those that will ultimately be obtained from the analyses after considering baseline seropositivity, clustering and attrition of participants.

Standard descriptive analyses will be performed to compare the characteristics of the study participants across sites. In general, missing data will not be imputed and either discarded or explicitly accounted for in the analyses. However, if more than 10% of data is missing for one or more key variables, the impact of missing data on the analysis will be discussed and potentially addressed using imputation methods.

Primary outcomes expressed as crude incidence rates will be calculated using standard person-time methods with person-time of follow-up as a denominator and 95% confidence intervals calculated using the Poisson distribution. For risk factors analyses, we will use Generalized Linear Mixed models, to account for potential confounders, with nested random effect on

communities and households to account for clustering of participants. Both univariate and multivariate models will be used to estimate adjusted incidence rate ratio (IRR).

Pre-defined interim analyses will be undertaken at the end of the first LF season, upon completion of the laboratory analyses of the samples. The main purpose of this interim analysis is to assess the need for continuation of the study through a second LF season of data collection, as well as to identify needs and inform plans for additional data collection or other activities beyond the original scope of the *Enable* programme. Final analyses will be carried out at the completion of the study.

We refer to the SAP (see supplement) for a complete description of the sample size calculation as well as the methods for interim and final analyses.

## Monitoring

Site monitoring visits will be conducted by MMARCRO for assessment, training, activation, interim monitoring, and close out. Monitoring activities prior to study start will include site assessment for readiness and training of site teams, generation of activation checklists for the sites, ensuring sites have all the required SOPs and other source documents, and submission and receipt of Ethics Committee approvals. Monitoring activities during the study will include review of consent forms, adherence to the study protocol and good epidemiological practice, documentation of study deviations, assessment of site management procedures, and source data review.

## Ethical and regulatory considerations

Ethical approval for the study will be sought from each national ethics committee of participating countries, from the institutional ethics committees where the studies will take place, and from the ethics committees of partnering institutions (see supplementary material).

## Discussion

The *Enable* programme is the largest and most comprehensive prospective cohort of LF to date and will generate crucial data on the epidemiology of LASV infection and LF disease in endemic countries in West Africa to support late-stage LF vaccine development informing feasibility, size, and location of LF vaccine clinical trials in West Africa. The *Enable* programme will provide a better understanding of the clinical spectrum of LF, how LASV infection and LF disease are related, the seasonality of the disease, the burden of SNHL among LF survivors, and the extent, if any, of seroreversion among LASV-positive individuals.

A key strength of the *Enable* programme is the close alignment of the partners and processes. This is being achieved through the strong support of CEPI, a robust governance structure, a PHQ working across all countries, strong leadership for each study, and the use of a core protocol. Development of a core protocol and standardised data collection tools with the input of all participating study country teams will ensure harmonisation of key aspects of the programme, such as inclusion and exclusion criteria, acute febrile case definition, study outcomes, and laboratory tests, which will maximise the comparability of data for between-country analyses. The subsequent adaptation by each country team to the context and resources of each study site will enable the study to be implemented in the most appropriate way to be acceptable to communities and participants, to maximise data quality, and to minimise loss to follow-up and under-reporting of febrile symptoms. Nevertheless, some adaptations and country-specific procedures may limit cross-country comparability. For example, each study country will compensate participants differently for their time at recruitment and follow-up, as

deemed appropriate in their communities, which may have a differential impact on the likelihood of households participating and retention rates.

Furthermore, the different contexts in each country are likely to impact study conduct and possibly, results. Political unrest in some of the countries might affect the conduct of the programme. The road infrastructure and distances over which field teams will travel for recruitment and follow-up activities and shipping of blood samples vary greatly, with the latter potentially affecting sample quality. Interruption to the power sources for freezers and lab equipment may affect sample storage and testing. Also, the availability of resources, such as the number of skilled and experienced research staff, laboratory capacity, and means of transport vary across study sites, which could affect data quality and study process timelines.

The active follow-up activities for the LF disease cohort will maximise the potential for detecting LF cases that may otherwise be missed through home treatment or misdiagnosis. However, the large sample sizes and long follow-up periods may mean follow-up activities and participant engagement will be difficult to maintain by the study teams. Also, those participants who drop out or are lost to follow-up will be likely to differ from those who remain in the study. For example, younger, healthier individuals may be more likely to travel out of the study area for work and be lost to follow-up or excluded. The country teams will make specific efforts to remain in contact with participants who migrate temporarily to facilitate study participation upon their return. The acceptability of giving blood samples for research purposes will require the field teams to conduct intense and ongoing community engagement and timely results reporting from tests. Willingness to continue giving samples may wane over time or if lab analyses get delayed, which could lead to missing data and underpowered analysis. Community engagement is an ongoing process and effective engagement is required to keep participants motivated.

Other infectious diseases have affected and likely will affect study conduct, and the study itself can facilitate outbreak response. The Ebola epidemic in West Africa between 2014–2016 continues to affect the healthcare-seeking behaviours of communities. In Sierra Leone, the same health facility for routine health services was used as the Ebola treatment centre and community members continue to be afraid to use the facility. During the Ebola outbreak in Guinea in 2021, CEPI was able to leverage an existing supply agreement to purchase Ebola test kits for use in *Enable* programme countries, capitalising on the partnership structure already in place to address needs for diseases other than Lassa. The COVID-19 pandemic has also impacted and will impact the study in several ways. Most noticeably, the start of the study was delayed by several months. Study staff will be affected directly through sickness; from the experience with Ebola there is likely to be community reluctance to visitors during recruitment; HCWs will be diverted to the COVID-19 response; coordinating partners will face difficulties during international travel to conduct training, monitoring, and field support; supply lines may be disrupted due to travel restrictions; and there are likely to be difficulties in procuring supplies such as lab consumables. Concern that individuals with LF symptoms will not seek care at health facilities owing to COVID-19 may affect the number of cases detected in our study [18]. The active follow-up of study participants in the community should overcome reluctance to attend health facilities in part. However, there remains the possibility that participants under-report symptoms at the time of follow-up to avoid being referred to health care for fear of being exposed to COVID-19.

At the end of the participant follow-up period, results will be disseminated through journal publications, conferences, national and regional health authorities, and to the communities themselves. The results of this programme will provide essential data to plan the location and scale of future late-stage trials of LF vaccine candidates.

## Supporting information

**S1 Fig. *Enable* programme governance structure.**
(TIFF)

## Acknowledgments

The *Enable* Protocol authorship group.

| | |
|---|---|
| Nnennaya Ajayi, Chiedozie Ojide, Emeka Onwe Ogah | Alex Ekwueme Federal University Teaching Hospital Abakaliki, No. 1 Chidume Street, P.M.B 102, Abakaliki, Ebonyi State, Nigeria |
| Elisabeth Fichet-Calvet | Department of Virology, Bernhard Nocht Institute for Tropical Medicine, Bernhard-Nocht-Straße 74, 20359, Hamburg, Germany |
| Alison A Bettis, Henshaw Mandi* | Coalition for Epidemic Preparedness Innovations (CEPI), Oslo, Norway |
| Gnoumè Camara, Alpha Mamoudou Diallo | Direction Préfectorale de la Santé, N1, Kissidougou, Faranah, Guinea |
| Chuks Abejegah, Nelson Adedosu | Owo Federal Medical Centre, Michael Adekun Ajasin Road, PMB, 1053, Owo, Ondo State, Nigeria |
| Olaniyi Abil Adeginka, Olouyomi Scherif Adegnika, Selidji Todagbe Agnandji, Akpeyedje Yanelle Dossou, Avokpaho Erupide, Josiane Honkpehendji, Francis Houeha, Mecit Issifou, Marie Alexandre Logbo | Fondation pour la Recherche Scientifique (FORS), 72 BP45 Cotonou, Bénin |
| Ekaete Tobin | Institute of Lassa Fever Research and Control, Irrua Specialist Teaching Hospital, KM 87 Benin Auchi Rd, Irrua, Edo State, Nigeria |
| Foday Alhassan, Simbirie Jalloh | Kenema Government Hospital (KGH), Kenema, Sierra Leone |
| Adejoke Akano, Elsie Ilori, Kamji Jan | Nigeria Centre for Disease Control, 801 Ebitu Ukiwe St, Jabi 900108, Abuja, Nigeria |
| Carol Kagia, Patrick Suykerbuyk | P95 Epidemiology and Pharmacovigilance, Koning Leopold Iii-Laan 1, Heverlee, Leuven 3001, Belgium |
| Andrea Bernasconi, Matthias Borchert | Robert Koch Institute, Nordufer 20, 13353 Berlin, Germany |
| Aissatou Bah, Mamadou Lamine Barry, Jacob Camara, Barrè Soropogui | Université Gamal Abdel Nasser de Conakry, Conakry, Guinea |

* Lead author of authorship group: Henshaw.mandi@cepi.net

We thank the following people for their roles in the setup of the *Enable* programme or contributions to content of this paper:

Ndapewa Ithete (BNITM), Paul Aloo, Gabrielle Breugelmans, Jakob Cramer, Roice Fulton, Gunnstein Norheim, Nicole Lurie, Beatriz Calvo Urbano, Solomon Yimer (CEPI), Boni Maxime Ale, Yves Amevoin, Patrick Barks, Paul Campbell, Marine Durthaler, Souna Garba, Mathilde Mousset, Mark Ndifon, Denis Opio, Eric Youm (Epicentre), Laleye Anatole, Dossou Ange, Rokiatou Babio, Ibrahim Mama Cissé, Moussa Djibril Mama Cissé, Gildas Hounkanrin, Adam Hounkpatin, Pélagie Mimonnou, Hountondji Michael, Gomina Moutakililou, Lydie Déré Chabi Nah, Jacob Namboni, Alassane Carine Tchibozo (FORS), Ibrahim Kapuwa (KGH), Amara Jambai, Sartie Kenneh, Thomas T Samba (Ministry of Health and Sanitation), Jacob Buanie (†), Mohamed Harding, Tiangay Kallon, Alhaji Jalloh, Mohamed Saio Kamara, (KGH Viral Haemorrhagic fever unit), Peter Smith (London School of Hygiene and Tropical Medicine), Elisabeth Diallo, Margaret Williams (MMARCRO), Ana Goios, David Hendrickx,

Archana Nagarajan, Jozica Skufca (P95), Nell G Bond, Emily J Engel, Crystal Zheng (Tulane University School of Medicine), McKenzie Colt, Sandrena Fischer, Alfred Flomo, Amara Fofana, Evelyn Gayflowu, Emmanuel Kerkula, Stanley Kerkula, Sam Livingstone, Abigail Morrison, Catherine Nimley, Kebeh Pewee, Carwolo Pewu, Thomas Remont, Alexander Sampson, Patience Segbee, Lonmine Sonah, Hanan Shuayto, Lonmine Sonah, Catherine Sumo, Thomas Sumo, Susan Sunay, George Tamba, Sam Tozay, Eleanor Watts, Zayzay Wolobah (UNC Project-Liberia, Phebe Hospital).

## Author Contributions

**Conceptualization:** Ayola Akim Adegnika, Danny Asogun, Olufemi Ayodeji, Benedict N. Azuogu, William A. Fischer, II, Robert F. Garry, Donald Samuel Grant, Magassouba N'Faly, Adebola Olayinka, Jefferson Sibley, David A. Wohl, Manfred Accrombessi, Chioma Dan-Nwafor, Rebecca Grais, Sylvanus Okogbenin, John S. Schieffelin, Thomas Verstraeten, Anges Yadouleton.

**Funding acquisition:** Ayola Akim Adegnika, Danny Asogun, Olufemi Ayodeji, Benedict N. Azuogu, Donald Samuel Grant, Christian Happi, Magassouba N'Faly, Adebola Olayinka, Robert Samuels, Jefferson Sibley, Manfred Accrombessi, Anton Camacho, Rebecca Grais, Stephan Günther, Chikwe Ihekweazu, Aminata Ndiaye, Robert Nsaibirni, Anges Yadouleton.

**Investigation:** Danny Asogun, Olufemi Ayodeji, Benedict N. Azuogu, Robert F. Garry, Donald Samuel Grant, Christian Happi, Adebola Olayinka, Robert Samuels, Chioma Dan-Nwafor, Akpénè Ruth Esperencia Deha, Augustine Goba, Énagnon Junior Juvénal Prince Honvou, Chikwe Ihekweazu, Lansana Kanneh, Mambu Momoh, Sylvanus Okogbenin, Ephraim Ogbaini, Énagnon Parsifal Marie Alexandre Logbo, John Demby Sandi, John S. Schieffelin, Anges Yadouleton, Emmanuel Koffi Yovo.

**Methodology:** Suzanne Penfold, Danny Asogun, Olufemi Ayodeji, Benedict N. Azuogu, William A. Fischer, II, Robert F. Garry, Donald Samuel Grant, Magassouba N'Faly, Adebola Olayinka, Robert Samuels, Jefferson Sibley, David A. Wohl, Manfred Accrombessi, Anton Camacho, Chioma Dan-Nwafor, Jean DeMarco, Sophie Duraffour, Augustine Goba, Rebecca Grais, Stephan Günther, Chikwe Ihekweazu, Christine Jacobsen, Lansana Kanneh, Mambu Momoh, Sylvanus Okogbenin, Ephraim Ogbaini, John Demby Sandi, John S. Schieffelin, Thomas Verstraeten, Emmanuel Koffi Yovo.

**Project administration:** Danny Asogun, Olufemi Ayodeji, Benedict N. Azuogu, William A. Fischer, II, Robert F. Garry, Donald Samuel Grant, Magassouba N'Faly, Adebola Olayinka, Robert Samuels, David A. Wohl, Ifedayo Adetifa, Giuditta Annibaldis, Chioma Dan-Nwafor, Akpénè Ruth Esperencia Deha, Jean DeMarco, Rebecca Grais, Énagnon Junior Juvénal Prince Honvou, Chikwe Ihekweazu, Christine Jacobsen, Sylvanus Okogbenin, Chinwe Ochu, Énagnon Parsifal Marie Alexandre Logbo, John S. Schieffelin, Nathalie J. Vielle.

**Resources:** Ifedayo Adetifa, Chikwe Ihekweazu.

**Software:** Anton Camacho, Aminata Ndiaye, Robert Nsaibirni.

**Supervision:** Ayola Akim Adegnika, Danny Asogun, Olufemi Ayodeji, Benedict N. Azuogu, William A. Fischer, II, Donald Samuel Grant, Christian Happi, Adebola Olayinka, Robert Samuels, Jefferson Sibley, David A. Wohl, Manfred Accrombessi, Ifedayo Adetifa, Giuditta Annibaldis, Chioma Dan-Nwafor, Jean DeMarco, Sophie Duraffour, Augustine Goba, Rebecca Grais, Stephan Günther, Chikwe Ihekweazu, Christine Jacobsen, Lansana Kanneh,

Mambu Momoh, Sylvanus Okogbenin, Chinwe Ochu, John Demby Sandi, Nathalie J. Vielle, Anges Yadouleton, Emmanuel Koffi Yovo.

**Validation:** Ayola Akim Adegnika, Donald Samuel Grant, Adebola Olayinka, Sophie Duraffour, Rebecca Grais, Stephan Günther, Christine Jacobsen, Anges Yadouleton.

**Visualization:** Anton Camacho, Aminata Ndiaye.

**Writing – original draft:** Suzanne Penfold.

**Writing – review & editing:** Ayola Akim Adegnika, Danny Asogun, Olufemi Ayodeji, Benedict N. Azuogu, William A. Fischer, II, Robert F. Garry, Donald Samuel Grant, Christian Happi, Magassouba N'Faly, Adebola Olayinka, Robert Samuels, Jefferson Sibley, David A. Wohl, Manfred Accrombessi, Ifedayo Adetifa, Giuditta Annibaldis, Anton Camacho, Chioma Dan-Nwafor, Akpénè Ruth Esperencia Deha, Jean DeMarco, Sophie Duraffour, Augustine Goba, Rebecca Grais, Stephan Günther, Énagnon Junior Juvénal Prince Honvou, Chikwe Ihekweazu, Christine Jacobsen, Lansana Kanneh, Mambu Momoh, Aminata Ndiaye, Robert Nsaibirni, Sylvanus Okogbenin, Chinwe Ochu, Ephraim Ogbaini, Énagnon Parsifal Marie Alexandre Logbo, John Demby Sandi, John S. Schieffelin, Thomas Verstraeten, Nathalie J. Vielle, Anges Yadouleton, Emmanuel Koffi Yovo.

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
