## [Decision Letter · Decision Letter 0]

18 Jan 2023

PONE-D-22-27085A Prospective Multi-Site Cohort Study to Estimate Incidence of Infection and Disease Due to Lassa Fever Virus in West African Countries (The Enable Lassa Research Programme) – Study ProtocolPLOS ONE

Dear Dr. Penfold,

Thank you for submitting your manuscript to PLOS ONE. After careful consideration, we feel that it has merit but does not fully meet PLOS ONE’s publication criteria as it currently stands. Therefore, we invite you to submit a revised version of the manuscript that addresses the points raised during the review process.

ACADEMIC EDITOR: Please address the comments raised by one of the reviewers and resubmit the manuscript.

We look forward to receiving your revised manuscript.

Kind regards,

Debdutta Bhattacharya

Academic Editor

PLOS ONE

Journal Requirements:

2. One of the noted authors is a group or consortium [insert name of group or team]. In addition to naming the author group, please list the individual authors and affiliations within this group in the acknowledgments section of your manuscript. Please also indicate clearly a lead author for this group along with a contact email address.

Reviewers' comments:

Reviewer's Responses to Questions

**Comments to the Author**

1. Does the manuscript provide a valid rationale for the proposed study, with clearly identified and justified research questions?

Reviewer #1: Yes

Reviewer #2: Yes

2. Is the protocol technically sound and planned in a manner that will lead to a meaningful outcome and allow testing the stated hypotheses?

Reviewer #1: Yes

Reviewer #2: Yes

3. Is the methodology feasible and described in sufficient detail to allow the work to be replicable?

Reviewer #1: Yes

Reviewer #2: Yes

4. Have the authors described where all data underlying the findings will be made available when the study is complete?

Reviewer #1: Yes

Reviewer #2: Yes

5. Is the manuscript presented in an intelligible fashion and written in standard English?

Reviewer #1: Yes

Reviewer #2: Yes

6. Review Comments to the Author

You may also provide optional suggestions and comments to authors that they might find helpful in planning their study.

Reviewer #1: 1. the study governance and roles of various partners - this part can be given in supplementary file

2. a map of the study sites can be given for better clarity

3. reason for including ">=2 years old" participants to be given

4. the inclusion criteria is fully not clear - are you going to recruit "healthy individuals" or LASV infection IgM/IgG positive individuals or LF disease patients (operational definition?)

5. This statement not clear: "draw the (smaller) LASV infection cohort as a nested cohort from the (larger, main) LF disease cohort" - shouldn't it be reverse as LASV infection is asymptomatic in approximately 80% of the cases. LF disease cohort should be nested in LASV infection cohort.

6. Clustering effect how to adjust for in sample size not mentioned

Reviewer #2: The manuscript is well written in a standard and easily readable fashion. Although the justification for the sample size does not seem adequate, the figure seem reasonably large for the desired precision. The methodology is clear, concise and very replicable. The study if optimally implemented will hopefully produce outcomes that will provide more insights into the Lassa fever situation in the most endemic regions in the world, thereby helping the control of the disease burden globally.

7. PLOS authors have the option to publish the peer review history of their article (what does this mean?). If published, this will include your full peer review and any attached files.

Reviewer #1: **Yes: **Tanveer Rehman

Reviewer #2: **Yes: **Mohammed Abdulkarim Abdullahi

---

## [Author Response · Author response to Decision Letter 0]

1 Feb 2023

Dear Drs. Chenette and Vousden, 

Thank you to both reviewers for reading our protocol papers. Please see our response to each of the points raised.

1. the study governance and roles of various partners - this part can be given in supplementary file

Response - This has been done

2. a map of the study sites can be given for better clarity

Response - This has been done

3. reason for including ">=2 years old" participants to be given

This was recommended by the local Lassa fever experts, who from their experiences, said that Lassa fever is very rare in the age group of less than two years. Also, community acceptability for blood draws from young children is low. Details on this rationale have been added to the inclusion/exclusion criteria on page 19 of the manuscript.

4. the inclusion criteria is fully not clear - are you going to recruit "healthy individuals" or LASV infection IgM/IgG positive individuals or LF disease patients (operational definition?)

Response - The text (page 19, line 5) has been amended to say ‘not febrile’ rather than healthy, which is more accurate and correct. This criterion was implemented to reduce the chance of recruiting active LF cases, but of course this would not exclude all with active infection as many are asymptomatic. It was not possible to conduct PCR tests on all participants at baseline.

5. This statement not clear: "draw the (smaller) LASV infection cohort as a nested cohort from the (larger, main) LF disease cohort" - shouldn't it be reverse as LASV infection is asymptomatic in approximately 80% of the cases. LF disease cohort should be nested in LASV infection cohort.

Response - Because the sample size required for the LASV infection cohort was smaller than that required for the LF disease cohort, nesting the disease cohort within the infection cohort was not possible. Lassa fever disease is the endpoint of future vaccine trials. Because it is unknown whether people can get Lassa fever disease more than once, baseline LASV seropositivity (IgG) did not preclude participation in the LF disease cohort.

6. Clustering effect how to adjust for in sample size not mentioned

Response – clustering considerations for the sample size calculation is mentioned in section ‘statistical analysis’, and is noted as a limitation as it was omitted: “The original sample size calculation in the protocol had several limitations: i) It could not account for the baseline LASV seroprevalence which is unknown in the study population. If the LASV seroprevalence in the study population is high, this is likely to reduce the number of people at risk in the cohort; ii) It did not take into account the clustering of participants within households and communities, which was necessary for feasibility and acceptance by the population; iii) It did not account for participant attrition during the 2-year follow-up period, which is likely to reduce the final cohort size for the analyses. As a result, the precision intervals specified in the protocol are likely to be narrower than those that will ultimately be obtained from the analyses after considering baseline seropositivity, clustering and attrition of participants.” As the other reviewer noted, “Although the justification for the sample size does not seem adequate, the figure seem(s) reasonably large for the desired precision.” 

We have also addressed the style requirements of the manuscript, figures (also processed using PACE), file naming, and authorship group. 

Yours sincerely, 

Suzanne Penfold

P95 Epidemiology and Pharmacovigilance 

Koning Leopold III laan 1 

3001 Leuven 

Belgium 

email: suzanne.penfold@p-95.com

---

## [Decision Letter · Decision Letter 1]

15 Mar 2023

A prospective, multi-site, cohort study to estimate the incidence of infection and disease due to Lassa fever virus in West African countries (the Enable Lassa research programme) - Study protocol

PONE-D-22-27085R1

Dear Dr. Penfold,

We’re pleased to inform you that your manuscript has been judged scientifically suitable for publication and will be formally accepted for publication once it meets all outstanding technical requirements.

Kind regards,

Debdutta Bhattacharya

Academic Editor

PLOS ONE

Additional Editor Comments (optional):

Reviewers' comments:

Reviewer's Responses to Questions

**Comments to the Author**

1. Does the manuscript provide a valid rationale for the proposed study, with clearly identified and justified research questions?

Reviewer #1: Yes

2. Is the protocol technically sound and planned in a manner that will lead to a meaningful outcome and allow testing the stated hypotheses?

Reviewer #1: Yes

3. Is the methodology feasible and described in sufficient detail to allow the work to be replicable?

Reviewer #1: Yes

4. Have the authors described where all data underlying the findings will be made available when the study is complete?

Reviewer #1: Yes

5. Is the manuscript presented in an intelligible fashion and written in standard English?

Reviewer #1: Yes

6. Review Comments to the Author

You may also provide optional suggestions and comments to authors that they might find helpful in planning their study.

Reviewer #1: Data on LASV infection and LF disease incidence in West Africa from this research programme will determine the feasibility of future Phase IIb or III clinical trials for LF vaccine candidates. All the comments have been addressed by the authors.

7. PLOS authors have the option to publish the peer review history of their article (what does this mean?). If published, this will include your full peer review and any attached files.

Reviewer #1: **Yes: **Tanveer Rehman

---

## [Editor Report · Acceptance letter]

20 Mar 2023

PONE-D-22-27085R1 

A prospective, multi-site, cohort study to estimate incidence of infection and disease due to Lassa fever virus in West African countries (the Enable Lassa research programme) – Study protocol 

Dear Dr. Penfold:

I'm pleased to inform you that your manuscript has been deemed suitable for publication in PLOS ONE. Congratulations! Your manuscript is now with our production department. 

Kind regards, 

on behalf of

Dr. Debdutta Bhattacharya 

Academic Editor

PLOS ONE